burnout; medical students; lifestyle; mental health; stress

**Corresponding author:**
Maria Helena Viegas da Cunha Gentil Martins;
Email: mariagentilviegas@gmail.com

# Burnout in medical students: A longitudinal study in a Portuguese medical school

Maria Helena Viegas da Cunha Gentil Martins[1] , Vasco Martins Lobo[1], Mafalda Sofia dos Santos Florenciano[1], Marco António Benjamim Morais[1] and Miguel Barbosa[2,3]

[1]Faculdade de Medicina, Universidade de Lisboa, Lisbon, Portugal; [2]CICPSI, Faculdade de Psicologia, Universidade de Lisboa, Lisbon, Portugal and [3]Instituto de Saúde Ambiental (ISAMB-FMUL), Faculdade de Medicina, Universidade de Lisboa, Lisbon, Portugal

## Abstract

**Background:** Burnout is highly prevalent among medical students. This study aimed to assess burnout levels over the course of a semester and identify variables that might explain burnout's variance over time.
**Method:** This longitudinal study involved medical students from a Portuguese school. Participants completed the Maslach Burnout Inventory and Mental Health Inventory-5, along with questions related to social support, help-seeking behaviours, academic performance, mental health and lifestyle assessment at the beginning (first phase), middle (second phase) and end (third phase) of the first semester of 2018–2019 academic year.
**Results:** A total of 108 participants provided responses in all phases (paired sample). The prevalence of burnout in the first phase was 28.2%, which increased to 34% in the second and 39.5% in the third. To explore factors contributing to burnout levels, we used the 332 responses obtained in the third phase (non-paired sample). Higher burnout levels were associated with poor academic performance, mental health stigma, consumption of tranquillisers and living away from home. Conversely, they were negatively associated with social support and a healthy lifestyle.
**Conclusions:** The study reveals a high prevalence of burnout among medical students, with burnout levels increasing throughout the semester. These levels are influenced by modifiable variables.

## Impact statement

This study was the first longitudinal study on burnout among medical students in Portugal. Our study provides a better understanding of the reality of burnout among medical students and the protective and negative variables related to it, shedding light on the alarming prevalence of burnout among medical students. By quantifying the widespread prevalence of burnout, we have validated the experiences of countless medical students who may have otherwise suffered in silence. This crucial data underscore the urgency of addressing burnout in medical education to safeguard the mental health and overall well-being of our future healthcare practitioners. Identifying the root causes of burnout has been a central achievement of our study. By elucidating the role of academic challenges, the need for a healthy lifestyle, with adequate time for sleep and physical activity as well as for social support, we have not only pinpointed areas of concern but have also provided empirical evidence to initiate structural changes within medical curricula. Our research holds significant implications for policy makers, medical educators and healthcare institutions alike. We advocate for a paradigm shift in medical education that promotes both academic excellence and personal well-being. The purpose of this study is to shed a light on the prevalence of burnout among medical students and guide the change, both academic and extracurricular, needed to decrease burnout's prevalence, and to advocate for better preventive strategies as well as for the growth and adaptation of the mental health services provided to the needs of their students.

## Introduction

Burnout is a syndrome characterised by emotional exhaustion, depersonalisation or cynicism and a decreased perception of personal accomplishment or reduced professional efficacy (Maslach et al., 1996; World Health Organization, 2019; Obregon et al., 2020). It has been extensively studied in various professional contexts. However, its impact on students, particularly those in demanding academic environments like medical schools, has gained increasing

attention. Recognising the need for a nuanced understanding of burnout in student populations, researchers have embarked on the task of adapting and renaming burnout dimensions to better align with the unique challenges and experiences faced by students.

A systematic review based on 24 studies that collectively assessed a total of 17,431 medical students worldwide found that 8,060 medical students experienced burnout, resulting in an estimated prevalence of 44.2% (Frajerman et al., 2019). Burnout among students typically initiates during medical school, persists beyond graduation and carries significant consequences for both patient care and the well-being of physicians. It is associated with increased cynicism towards patients, poorer academic results, thoughts of dropping out of medical school and professional misconduct (IsHak et al., 2013; Erschens et al., 2018; Mian et al., 2018; Jumat et al., 2020; Obregon et al., 2020; Alves et al., 2022). Moreover, mental health is believed to deteriorate progressively with each successive year of medical training (van Venrooij et al., 2015; Erschens et al., 2018). These findings suggest that the educational process itself may contribute to increased psychological distress (Erschens et al., 2018).

A poor lifestyle, psychiatric comorbidities and financial stress are also believed to contribute to increased levels of burnout (IsHak et al., 2013; Cecil et al., 2014; van Venrooij et al., 2015). However, only a few studies have explored the relationship between various lifestyle factors and burnout in medical students (Cecil et al., 2014; Lee et al., 2020). Conversely, strong social support networks and positive mental health serve as protective factors against burnout, helping students navigate the challenges of medical education more effectively (Alves et al., 2022).

At an institutional level, curriculum evaluations often neglect to assess its impact on students' mental well-being (Lee et al., 2020). Information regarding burnout among medical students can help medical schools in designing customised interventions, raising awareness about burnout and establishing baseline data for monitoring the efficacy of preventive measures (IsHak et al., 2013; Lee et al., 2020).

Our primary goal was to study the prevalence of burnout among medical students and analyse its variations during three phases of the first semester. The secondary aim was to examine the associations between burnout and demographic characteristics, lifestyle, economic stressors, social support, substance use, perception of mental health stigma, mental health and help-seeking behaviour.

## Methods

### Participants

The study's population consisted of all medical students at the Faculdade de Medicina da Universidade de Lisboa, which had a total of 2,160 students in the academic year 2018–2019. Among these students, 66.1% were female (Faculdade de Medicina da Universidade de Lisboa, 2019).

Participants who completed the first, second and third phases formed the paired sample. Additionally, a non-paired sample was derived from those who participated only in the third phase.

### Procedure

This prospective, longitudinal, observational study was conducted from September 2018 to January 2019. Students completed an online survey during three 18-day phases within the first semester: baseline phase (from 19/09/2018 to 07/10/2018) at the beginning of academic year; middle-of-semester phase (14/11/2018 to 02/12/2018); and pre-exam season and first week of exams phase (01/01/2019 to 18/01/2019). Students were informed that participation was voluntary, and all provided information would remain confidential and anonymous. They provided consent for the use of their data.

The study was conducted online through a Google Form that did not collect emails. The survey link was shared with all medical students via institutional email.

### Measures

#### Demographics and clinical measures

We gathered demographic information including age, gender, academic year, academic statute, economic stress (on a 5-point Likert scale) and living arrangements (whether students lived at home with their family or not). For those living away from home, we used a 5-point Likert scale to determine how often they returned home each month.

#### Academic performance

Academic performance was assessed by asking how many subjects the student had to repeat in the current academic year and whether they had failed a year.

#### Burnout

Students' burnout was assessed by the Portuguese version of the Maslach Burnout Inventory – Student Survey (Schaufeli et al., 2002), which was validated by Maroco and Tecedeiro (2009). The MBI-SS is a self-report questionnaire containing 15 items, each rated on a 7-point Likert scale and grouped into three dimensions: exhaustion (five items; α = 0.91); cynicism (four items; α = 0.92) and efficacy (six items; α = 0.85). The exhaustion, cynicism and efficacy scores are calculated by summing the respective items. To derive a global burnout score, the scores from all three dimensions were averaged. Additionally, an individual was categorised as experiencing burnout, relative to their group, if they scored above the 66th percentile on the exhaustion and cynicism while scoring below the 33rd percentile on efficacy (Maroco and Tecedeiro, 2009).

#### Social support

Social support was assessed by asking whether students were in a romantic relationship and if they perceived it as healthy, using a 5-point Likert scale.

Additionally, social satisfaction was assessed using a revised version of the Satisfaction with Social Support scale (Ribeiro, 1999), comprising three items rated on a 5-point Likert scale: "I am satisfied with the way I relate to my family", "If I need emergency support, I have several people that I can turn to" and "When I need to vent to someone, I easily find support from my peers" (α = 0.74). The final score was calculated as the mean of these three items.

#### Mental health

Mental health status was determined by asking students if they had received a diagnosis of any mental health condition and whether they were currently receiving treatment or support for it.

Additionally, the Portuguese version (Ribeiro, 2001) of the Mental Health Inventory-5 (MHI-5) was used to assess students' mental health. The MHI-5 is a reduced version of the 38-item Mental Health Inventory (Veit and Ware, 1983) and consists of five items assessed using a 6-point Likert scale (α = 0.90).

**Table 1.** Sociodemographic characteristics of participants in the three phases (non-paired samples) and in the paired sample

| | P1 | | P2 | | P3 | | Paired sample | |
|---|---|---|---|---|---|---|---|---|
| | n | % | N | % | n | % | N | % |
| Gender | | | | | | | | |
| Female | 329 | 74.3 | 302 | 74.9 | 264 | 79.5 | 85 | 78.7 |
| Male | 114 | 25.7 | 101 | 25.1 | 68 | 20.5 | 23 | 21.3 |
| Other | 0 | 0 | 0 | 0 | 0 | 0 | 0 | 0 |
| Academic year | | | | | | | | |
| 1 | 69 | 15.6 | 75 | 18.6 | 83 | 25 | 16 | 14.8 |
| 2 | 110 | 24.8 | 99 | 24.6 | 79 | 23.8 | 34 | 31.5 |
| 3 | 119 | 26.9 | 90 | 22.3 | 76 | 22.9 | 32 | 29.6 |
| 4 | 68 | 15.3 | 59 | 14.6 | 45 | 13.6 | 16 | 14.8 |
| 5 | 34 | 7.7 | 53 | 13.2 | 32 | 9.6 | 7 | 6.5 |
| 6 | 43 | 9.7 | 27 | 6.7 | 17 | 5.1 | 3 | 2.8 |
| Academic special status | | | | | | | | |
| None | 423 | 95.5 | 392 | 97.3 | 316 | 95.2 | 105 | 97.2 |
| With status | 20 | 4.5 | 11 | 2.7 | 16 | 4.8 | 3 | 2.8 |
| Place of living | | | | | | | | |
| Outside of home | 261 | 58.9 | 222 | 55.1 | 187 | 56.3 | 60 | 55.6 |
| Home | 182 | 41.1 | 181 | 44.9 | 145 | 43.7 | 48 | 44.4 |

*Perceived stigma and help-seeking behaviour of medical students*

To assess help-seeking behaviour, we examined students' awareness of the mental health support services offered by the faculty. Students were asked if they were aware of the existence of these services and whether they had used them.

A revised version of the Survey about Stigma and the Help-Seeking Behaviours of Medical Students with Burnout (Dyrbye et al., 2015) was adapted for the Portuguese context to assess perceived and experienced mental health stigma among both peers and faculty. This self-report questionnaire consists of five items (specifically, items 1, 2, 3, 4 and 10 from the original survey), each rated on a 5-point Likert scale. The scale is unidimensional ($\alpha = 0.81$), and scores for perceived mental health stigma were derived by calculating the average of these five items.

*Lifestyle*

Dietary habits were evaluated using a 5-point Likert scale to assess the regularity of meals. Physical activity frequency was determined by asking how often students engaged in physical activity per week. We also assessed sleep duration (in hours) over the past month and its perceived adequacy for academic performance using a 5-point Likert scale. Study habits were assessed by asking about daily study hours and whether students considered it sufficient for their academic performance using a 5-point Likert scale.

Regarding extracurricular activities, they reported their average weekly hours and their perception of these activities as stress-reducing using a 5-point Likert scale.

*Substance use*

Substance use was assessed by asking students how many times a week they had used substances in the last month using a 9-point Likert scale. The included substances were tobacco, tranquillisers, sleep medication, stimulant medication, recreational drugs and non-prescribed cognitive stimulants.

*Data analysis*

Descriptive statistics were used to describe the sociodemographic characteristics.

*Paired sample*

In the paired sample, burnout levels in the three phases were assessed to examine the progression over the semester and to evaluate burnout levels among students in each academic year during each phase. An ANOVA was used to determine variations in burnout levels across the three different phases in all three categories. A General Linear Model – Repeated Measures analysis was performed to assess how burnout varied according to participants' academic year and the three phases. Mauchly's test was used to assess sphericity.

*Non-paired sample*

An ANOVA was performed to assess the differences in burnout levels among the three phases.

To investigate the correlation between burnout levels and the remaining study variables, we select the third phase sample (the non-paired sample) due to its highest burnout levels and proximity to the exam season.

A *t*-test was performed to assess differences in burnout during the third phase on categorical variables, including gender, academic statute, living arrangements, romantic relationship status, awareness of the faculty's mental health support service and previous usage.

Pearson correlations were performed to assess the magnitude of the association between total burnout levels in the third phase and

**Table 2.** Frequencies and percentages of burnout in the three phases

|         | Without burnout | Burnout |
|---------|-----------------|---------|
| Phase 1 | 318 (71.8%) 1.3 | 125 (28.2%) −1.9 |
| Phase 2 | 266 (66%) −0.2 | 137 (34%) 0.2 |
| Phase 3 | 201 (60.5%) −1.4 | 131 (39.5%) 1.9 |

*Note*: Each cell contains the frequency, percentage and standardised residuals; Pearson chi-square = 10.90, SD = 2.00, $p$ = 0.004.

the following variables: age, academic year, economic stress, monthly frequency of returning home, number of failed academic subjects, failing a school year, healthy romantic relationship status, social support, overall mental health, perception of mental health stigma, dietary habits, physical activity, sleep habits, study habits, weekly time dedicated to extracurricular activities, and their relationship to self-perceived stress levels, as well as substance use.

A multiple linear regression analysis was performed to predict burnout in the third phase.

All statistical analyses were conducted using SPSS 27.0, with statistical significance set at $p < 0.050$.

## Results

### Participants

Out of the 2,160 medical students at Faculdade de Medicina da Universidade de Lisboa, 443 responses (20.5%) were received in the first phase, 403 (18.66%) in the second phase and 332 (15.37%) in the third phase.

A total of 108 participants completed all three phases, constituting 5% of the enrolled students, which formed the paired sample. In this paired sample, the respondents had an average age of 20.32 years ($SD = 1.87$ | range: 17–27 years).

### Psychiatric disorders

In the first phase, the prevalence of previously diagnosed psychiatric disorder was 17.6%, followed by 15.6% in the second phase and 14.8% in the third phase. The majority of this group received some form of support, such as medication, psychotherapy, or both. However, a significant portion of them were not receiving any support for their condition (P1 = 24.7%; P2 = 31.1%; P3 = 32.8%).

### Burnout prevalence and progression during the semester

Table 2 presents the burnout prevalence in each phase.

Burnout's prevalence was 28.2% in the first phase, 34% in the second phase and 39.5% in the third phase.

The results of the ANOVA assessing differences in burnout levels across the three phases are shown in Table 3.

Burnout levels significantly increased in the second phase compared to the first phase and further increased in the third phase compared to both the first and second phases.

Figure 1 shows the evolution of burnout across the three phases in the paired sample by academic year.

First-year students initially exhibit lower burnout levels compared to other students but experience the most significant increase in burnout from the first to the second phase. In contrast, sixth-year medical students consistently demonstrate higher levels of burnout

across all study phases. Mauchly's test of sphericity showed that this assumption was met: $\chi^2(2) = 5.85$, $p = 0.054$.

There was a significant moderate effect of the study phase on burnout ($F(2,204) = 7.23$, $p = 0.001$, $\eta_p^2 = 0.066$). When comparing mean differences between the three phases in different academic years, only first-year students exhibited significant differences between the first phase ($M = 2.10$, $SD = 2.72$), second phase ($M = 2.72$, $SD = 1.06$) and third phase ($M = 2.79$, $SD = 1.22$) ($F(2,30) = 9.36$, $p = 0.001$, $\eta_p^2 = 0.38$). For students in other academic years, there were no significant differences in burnout levels between the three phases.

In the first phase, significant differences between first-year students ($M = 2.10$; $SD = 0.80$) and second-year students ($M = 2.71$; $SD = 0.97$) ($F(5,203) = 3.00$, $p = 0.013$, $\eta_p^2 = 0.07$), as well as first-year students ($M = 2.10$; $SD = 0.80$) and sixth-year students ($M = 3.24$; $SD = 0.72$) ($F(2,203) = 3.00$, $p = 0.041$, $\eta_p^2 = 0.07$).

In the second and third phases, no significant differences were observed in burnout levels on the academic year of the students.

### Correlations and differences in total burnout level and other study variables

There were no significant differences between the total burnout level and the following variables: age, year of study, monthly frequency of returning home, self-perceived healthy romantic relationship, daily study hours, weekly hours dedicated to extracurricular activities, perceived stress reduction from extracurricular activities, and the use of alcohol, tobacco, sleep medication, stimulant medication, recreational drugs and non-prescribed stimulant substances.

Significant differences were found between the total burnout level and the following variables: economic stress ($r = 0.17$, $p = 0.002$), failing a school year ($r = 0.27$, $p < 0.001$), number of failed academic subjects ($r = 0.25$, $p < 0.001$), social support ($r = -0.36$, $p < 0.001$), overall mental health ($r = -0.62$, $p < 0.001$), perception of mental health stigma ($r = 0.19$, $p < 0.001$), dietary habits ($r = -0.27$, $p < 0.001$), physical activity ($r = -0.19$, $p = 0.001$), average hours of sleep per day ($r = -0.23$, $p < 0.001$), perceived adequate sleep for academic performance ($r = -0.35$, $p < 0.001$) and the use of tranquillisers ($r = 0.22$, $p < 0.001$).

Additionally, the average hours of sleep per day positively correlated with the students' perception of sleep adequacy for academic performance ($r = 0.54$, $p < 0.001$).

Total burnout levels among students living away from home ($M = 2.93$; $SD = 0.99$) were significantly higher than those among students living at home while studying at faculty ($M = 2.71$; $SD = 1.00$) ($t(330) = 2.01$, $p = 0.045$).

### Predictors of burnout in the third phase sample

Table 4 reports the results of the multiple linear regression analysis aimed at identifying burnout-predicting variables. We included variables that exhibited a significant association with burnout in this analysis.

Social support and academic failure were the variables that showed the most significant association with burnout, with social support exerting an inverse impact on burnout levels.

Sleep and dietary habits, along with weekly physical activity, showed a negative association with burnout, while living away from home and the perception of mental health stigma were correlated to higher burnout levels.

**Table 3.** Means, standard deviations and analysis of variance (ANOVA) results for burnout

| | Phase 1 | | Phase 2 | | Phase 3 | | |
|---|---|---|---|---|---|---|---|
| | M | SD | M | SD | M | SD | F(2,1175) |
| Total burnout | 2.45[a] | 1.00 | 2.65[b] | .96 | 2.83[c] | 1.00 | 14.53*** |
| Exhaustion | 2.79[a] | 1.21 | 2.99[b] | 1.22 | 3.28[c] | 1.29 | 14.92*** |
| Cynicism | 1.62[a] | 1.46 | 1.96[b] | 1.52 | 2.13[b] | 1.56 | 11.69*** |
| Efficacy | 2.72[a] | 1.02 | 2.82 | 1.00 | 2.93[b] | 0.99 | 4.15** |

*Note*: The superscript alphabets "a, b and c" indicate that there are significant differences between each phase's means. Means sharing a common superscript are not statistically different at $p < 0.050$, according to the Tukey HSD procedure.***$p < 0.001$.
**$p < 0.050$.

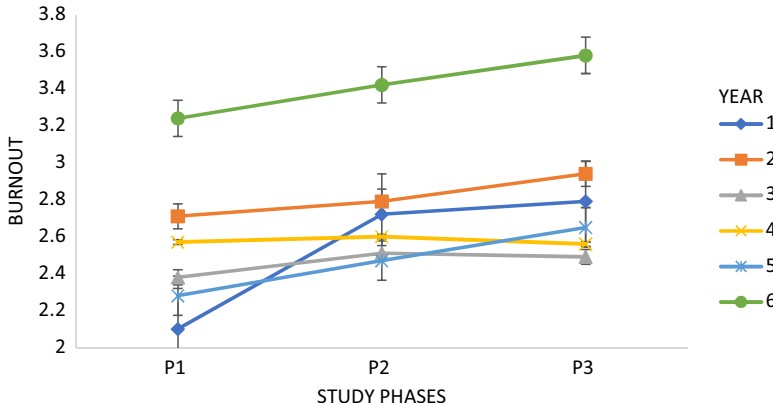

**Figure 1.** Evolution of burnout across the three phases in the paired sample by academic year. *Note*: Mean and standard error of the first phase: 1st year $M = 2.10 \pm 0.14$ (SE); 2nd year $M = 2.71 \pm 0.09$ (SE); 3rd year $M = 2.38 \pm 0.08$ (SE); 4th year $M = 2.57 \pm 0.12$ (SE); 5th year $M = 2.28 \pm 0.20$ (SE); 6th year $M = 3.24 \pm 0.15$ (SE). Second phase: 1st year $M = 2.72 \pm 0.12$ (SE); 2nd year $M = 2.79 \pm 0.09$ (SE); 3rd year $M = 2.47 \pm 0.09$ (SE); 4th year $M = 2.60 \pm 0.14$ (SE); 5th year $M = 2.47 \pm 0.14$ (SE); 6th year $M = 3.42 \pm 0.17$ (SE). Third phase: 1st year $M = 2.79 \pm 0.11$ (SE); 2nd year $M = 2.94 \pm 0.11$ (SE); 3rd year $M = 2.49 \pm 0.11$ (SE); 4th year $M = 2.56 \pm 0.13$ (SE); 5th year $M = 2.49 \pm 0.21$ (SE); 6th year $M = 3.58 \pm 0.21$ (SE).

## Discussion

This study investigated burnout prevalence in medical students and its progression throughout the first semester using a paired sample. It also explored the association between burnout levels in the third phase and various variables, including demographic characteristics, lifestyle, economic stressors, social support, substance use, perception of mental health stigma, mental health and students' help-seeking behaviour.

### Prevalence of the burnout

Our findings reveal that burnout affected 28.2% of students in the first phase, which notably escalated to 34% in the second phase and 39.5% in the third phase. These burnout levels are consistent with previous research, aligning with an estimated prevalence of 44.2% [33.4%–55.0%] (Frajerman et al., 2019).

A noteworthy increase in burnout levels was observed between the first and second phases, particularly among first-year medical students. In the first phase, they exhibited the lowest burnout levels but rose to the third-highest position by the second phase. This trend may align with the documented higher prevalence of burnout in medical students compared to their non-medical counterparts (Shanafelt et al., 2012). Conversely, sixth-year medical students consistently showed the highest burnout levels among all students, highlighting that those nearing the end of their studies tend to experience more significant mental health challenges than their peers (Liu et al., 2018).

### Predictors of burnout in medical students

Furthermore, this study has identified demographic and lifestyle factors that may serve as predictors of burnout in medical students. These factors include economic stress, dietary habits, physical activity, sleep habits and residing away from home. Additionally, other variables, such as academic performance, social support, overall mental health, perception of mental health stigma and the use of tranquillisers, have also been associated with burnout.

Higher levels of burnout were positively associated with factors such as academic year failure, the number of failed academic subjects, perception of mental health stigma, use of tranquillisers and residing away from home while studying. These findings are consistent with previous research (Dyrbye et al., 2015; Mian et al., 2018; D'Alva-Teixeira et al., 2023). Notably, academic year failure exhibited the strongest correlation with increased burnout levels, thus demonstrating the impact of academic performance on medical students' mental health. Additionally, incomplete subjects from previous academic years are added to the burden, contributing to higher burnout levels. It is worth emphasising that burnout can also adversely affect academic performance, potentially creating a reciprocal relationship between burnout and poor academic outcomes (Shadid et al., 2020). Therefore, proactive support measures should be implemented, particularly for students struggling with the course failures.

Contradictory findings on the use of tranquillisers have been reported in various studies (Erschens et al., 2018; Andrade et al., 2021). It is possible that students who were already taking

**Table 4.** Regression coefficients for predicting burnout levels in phase 3

| Variable | B | 95% CI | β | t | p |
|---|---|---|---|---|---|
| (Constant) | 4.304 | [3.183, 5.426] | | 7.55 | <0.001 |
| Economic stress | 0.008 | [−0.075, 0.095] | 0.009 | 0.181 | 0.856 |
| Failing a year | 0.750 | [0.333, 1.168] | 0.176 | 3.533 | <0.001 |
| Social support | −0.285 | [−0.416, −0.155] | −0.231 | −4.296 | <0.001 |
| Living away from home | 0.212 | [0.013, 0.410] | 0.106 | 2.094 | 0.037 |
| Physical activity per week | −0.109 | [−0.224, 0.006] | −0.093 | −1.86 | 0.063 |
| Dietary habits | −0.114 | [−0.209, −0.020] | −0.124 | −2.378 | 0.018 |
| Sleep habits | −0.162 | [−0.257, −0.068] | −0.167 | −3.367 | 0.001 |
| Perception of mental health stigma | 0.131 | [0.013, 0.248] | 0.110 | 2.193 | 0.029 |

*Note*: $R^2_{adj}$ = 0.25 ($N$ = 332, $p$ < 0.001. CI, confidence interval for $B$.

tranquillisers at the beginning of this study were experiencing burnout or dealing with underlying mental health issues that put them at greater risk of burnout. Future research should delve deeper into this variable to provide a clearer understanding.

Higher levels of burnout were negatively associated with social support, physical activity, self-perceived dietary adequacy and adequate sleep habits. These findings align with prior research, emphasising the necessity of promoting healthy lifestyles among medical students (Macilwraith and Bennett, 2018; Mian et al., 2018; Dinis et al., 2020; Lee et al., 2020). Social support exhibited the strongest negative correlation with burnout, highlighting its protective role. Implementing mentoring programmes that offer peer support and structured wellness initiatives to enhance stress management skills can be effective strategies to reduce burnout in medical students (Dyrbye et al., 2005; IsHak et al., 2013; West et al., 2016; Mian et al., 2018). Given the increased burnout risk for students living away from home, medical schools should also implement support measures for this specific group.

The importance of nurturing a social support network and promoting healthy lifestyles is crucial in mitigating burnout (Mian et al., 2018; Alves et al., 2022). It is highly advisable to implement interventions aimed at nurturing wellness during medical training. These interventions include creating awareness and conducive environments for adopting healthy lifestyles (IsHak et al., 2013; Mian et al., 2018; Lee et al., 2020). Medical schools should actively endorse healthy living throughout the academic journey and play a proactive role in advocating for healthy habits. This necessitates comprehensive planning within the medical curriculum to accommodate a healthy lifestyle (Mian et al., 2018; Dinis et al., 2020). Emphasising self-care skills and facilitating adequate sleep routines should be integral components of curriculum adaptation (Frajerman et al., 2019).

Mental health stigma delays help-seeking and medical students often postpone seeking mental health care when experiencing burnout, resulting in higher burnout levels (Dyrbye et al., 2015). Given the correlation between the perception of mental health stigma and burnout, it is essential to prioritise efforts to combat this stigma. This can be achieved through advocacy, education about mental health and psychological support, along with better promotion of the mental health services offered by the faculty (Dyrbye et al., 2015).

For the students already experiencing symptoms, there is a need for effective intervention programmes (Dyrbye et al., 2005; Frajerman et al., 2019). Counselling services should be actively promoted and made accessible to medical students, either at no cost or at a reduced and affordable fee, to ensure that economic barriers do not hinder access to necessary mental healthcare. These services should also be provided promptly.

### Limitations and future studies

Only 5% of students at the Faculdade de Medicina da Universidade de Lisboa completed all three survey phases, which may have influenced the observed burnout levels among medical students and the choice of statistical analysis. We recognised the limitations of using the generalised linear model analysis instead of structural equation modelling for assessing the impact of variables on burnout levels across the study's phases.

Additionally, there is an uneven representation of fifth-year and sixth-year medical students in the sample compared to students from other academic years.

Future research should prioritise further investigation into the correlation between medical schools' well-being promotion strategies and their effects on reducing burnout among medical students. Additionally, exploring the association between burnout and lifestyle factors, an area with limited existing research, is essential. Finally, future studies should aim to uncover reasons behind the evolution of burnout as students' progress through their medical studies, ideally by following students consistently from the first to the sixth year of medical education.

### Conclusions

This study highlights the high levels of burnout among medical students, which tend to increase as the demands of medical school intensify. The findings regarding the progression of burnout levels over time and their associated factors offer valuable insights for designing burnout prevention strategies.

This study underscores the importance of protective factors such as higher social support, regular physical activity, self-perceived healthy dietary habits and sufficient sleep. These factors align with existing literature, with social support showing the strongest inverse correlation with burnout. These factors align with existing literature, with social support showing the strongest inverse correlation with burnout promoting both a healthy lifestyle and effective support systems in preventing burnout.

We recommend that medical schools prioritise the creation of a more supportive and healthier learning environment. This includes

initiatives to combat mental health stigma and providing counselling and mentoring programmes to support their students.

**Open peer review.** To view the open peer review materials for this article, please visit http://doi.org/10.1017/gmh.2023.61.

**Data availability statement.** The data that support the findings of this study are available from the corresponding author, M.H.V.C.G.M., upon reasonable request.

**Author contributions.** All authors contributed to the conceptualisation, design and the collection of data for this study. M.H.V.C.G.M., V.M.L. and M.B. did the statistical analysis and interpretation of data. M.H.V.C.G.M. and M.B. wrote the main manuscript text and revised it.

**Financial support.** This work was supported by the Faculdade de Medicina da Universidade de Lisboa Office of Support to Scientific, Technological and Innovation Investigation (GAPIC) through the 24th Programme "Education through Science" directed at University of Lisbon School of Medicine students (Grant No. 20210011).

**Competing interest.** The authors do not have conflicts of interest.

**Ethics statement.** This research study was conducted in accordance with the ethical principles outlined in the Declaration of Helsinki. Informed consent was obtained from all participants after receiving detailed information about the study's purpose, procedures, risks and benefits. Participants voluntarily agreed to participate, and they were assured of their right to withdraw from the study at any point without consequences. Stringent measures were taken to safeguard participant privacy and data confidentiality, including anonymisation and secure storage. Participant identities remained confidential in all research outputs.

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
