## [Reviewer Report]

Dear editorial board

The article we will be submitting to your journal, titled “Burnout in Medical Students: A Longitudinal Study in Lisbon’s School of Medicine”, is a longitudinal, prospective and observational study, conducted among medical students at Lisbon’s School of Medicine. 

Given that Burnout is highly prevalent among medical students and represents a challenge to their well-being, health and academic success, and consequentially to their patients, specially given that without proper care our students will be burned-out doctors with all the consequences associated with a poor mental health in patient care, we felt it was essential to know our reality in terms of burnout prevalence, variation through a semester and possible influencing variables.

Prior to this study, there were no longitudinal studies of burnout in medical students in Portugal. Our primary goal was to assess the prevalence of burnout in medical students in Lisbon’s School of Medicine and how it varied through the first semester, using the beginning of the semester as our baseline and assessing variations between school years.

Our secondary goal was to determine the relationship between demographic characteristics, lifestyle, economic stressors, social support, substance use, perception of mental health stigma, mental health, help-seeking behaviour, and burnout levels in the phase with the highest burnout level. 

We believe it is essential to know our reality in order to implement meaningful change and effective prevention programs. By recognizing we have a problem we are able to work towards a solution. By knowing what influences the burnout levels of our students, we are able to provide information and knowledge for the development of preventive strategies and provide adequate mental health support for those already affected and those at higher risk. This study was developed first and foremost to be useful in changing our reality and promote prevention. 

Given that your journal is a peer reviewed journal dedicated to mental health, which encompasses burnout, and given our article provides information on possible interventions to prevent burnout and reduce risk factors, and advocates for the promotion of mental health, we believe our article fits into the purpose and scope of your journal.

We would like to thank you for your time and consideration. 

Sincerely,

Maria Gentil Viegas, M.D

---

## [Reviewer Report]

There is a very timely and useful article and it addresses a real concern in medical education. The findings are very similar to what is expected based on available literature. I have no issue with the substance of the article, but I hereby recommend the following changes.

For the Method and the results, I propose that you delete the entire section that is related to the paired sample. In my opinion, the non-paired sample is enough. Also it would be helpful if r values could have been presented in a tabular form as this will allow for easier reading

The discussion is well written and contains all of the elements according to the STROBE guidelines and the conclusions presented are supported by the results, however it is my opinion that there should be a line about the role of the sociodemographic factors that influence burnout in the conclusions

There is a graph that examines burnout by study phases and year, however it was not clear that it was the year you were referring to and therefore a minor annotation on this graph would be useful.

1. The introduction contains provides a good context and lays the foundation for the rest of the study, however a complete rewrite of the introduction in my opinion is needed so that it sounds like a single coherent piece. As it currently stands, the introduction appears to be an assemblage of sentences that do not connect well to each other.

---

## [Reviewer Report]

The study shows potential interest, but it has significant methodological limitations and flaws that undermine the validity of the results and conclusions. The major issues are as follows:

1. The literature review is limited, neglecting previous studies on Burnout in Portuguese Student Populations, including a recent (2021) study conducted with the same study population (medical students at the Faculdade de Medicina, University of Lisbon). The adaptation and renaming of the Burnout dimensions for the student population should be briefly explained. It is important to note that Student Burnout is not the same as general burnout in aid professions.The same applies to the selection of other variables. The rationale behind their inclusion (e.g., Social Support or Mental Health) and the evidence that supports their role as precursors of burnout should be provided. Additionally, it should be acknowledged that there are two medical schools in Lisbon.

2. The psychometric properties of the latent measures used (Burnout, Social Support, Mental Health) are not adequately described. Were the data collected in the present sample gathered using valid and reliable scales?

3. The major study variables are latent, so Structural Equation Modeling should be employed instead of classical repeated measures ANOVA and correlational analysis. It is also advisable for the authors to reconsider the use of the “A-NOVA” abbreviation, as it is not commonly used. With three time points, I would recommend using latent growth models.

4. The sample used in the study is not representative, as only 5% of the study population was selected, and the selection was not random. Consequently, the external validity of the conclusions is limited. Resorting to classical Generalized Linear Model (GLM) analysis, instead of Structural Equation Modelling, as the authors acknowledge, is not the solution.

5. It is important to clarify the cutoff values for the Burnout scores that were considered indicative of Burnout. How were the Burnout scores calculated? Who validated the Portuguese version of the Maslach Burnout Inventory – Student Survey? What evidence of validity and reliability exists for the MBI-SS in the Portuguese population? The same questions apply to the other scales used. The manuscript lacks important information in this regard.

6. Provide standard errors (SE) for the data points in Figure 1.

I regret to inform you that, after this thorough review, the outcome is not positive, particularly considering that this manuscript represents a students' research effort on an important subject. While the work has potential, it falls short of the standard required for publication in a journal like Cambridge Prisms.

---

## [Reviewer Report]

Dear Dr. Judith Bass,

Thank you for considering our revised manuscript titled “Burnout in Medical Students: A Longitudinal Study in Lisbon School of Medicine” for submission to the Cambridge Prisms: Global Mental Health. We appreciate the time and effort invested by you and the reviewers in providing valuable feedback on our work. We are also appreciative of the reviewers‘ insightful comments on our manuscript. Regarding the reviewers’ suggestions, we have diligently incorporated changes that address the majority of their recommendations. Below, we provide a point-by-point response to the comments and concerns raised by the reviewers.

We have changed the name of the manuscript to “Burnout in medical students: A longitudinal study in a Portuguese medical school”. Our Impact statement has also been reviewed. The paper has been completely reviewed.

REVIEWER 1:

1. “For the Method and the results, I propose that you delete the entire section that is related to the paired sample. In my opinion, the non-paired sample is enough”: 

Response: Thank you for your recommendations to improve the paper’s clarity. However, the paired sample was essential for assessing how burnout levels varied during the three phases of our longitudinal study. It played a crucial role in achieving one of our main objectives, which is to evaluate how burnout changes over the course of the semester. Without the paired sample, this analysis wouldn’t have been possible. 

2. “Also it would be helpful if r values could have been presented in a tabular form as this will allow for easier reading”

Response: We agree that presenting these results in a table would be clearer. However, due to the large number of variables and their correlations with each other, as well as the limit of 5 tables set by the journal, it was necessary to adapt the way we present the results.

3. “The discussion is well written and contains all of the elements according to the STROBE guidelines and the conclusions presented are supported by the results, however it is my opinion that there should be a line about the role of the sociodemographic factors that influence burnout in the conclusions. There is a graph that examines burnout by study phases and year, however it was not clear that it was the year you were referring to and therefore a minor annotation on this graph would be useful. The introduction provides a good context and lays the foundation for the rest of the study, however a complete rewrite of the introduction in my opinion is needed so that it sounds like a single coherent piece. As it currently stands, the introduction appears to be an assemblage of sentences that do not connect well to each other.”

Response: The suggestions regarding the introduction, Figure 1, and conclusion have been integrated into the new version.

REVIEWER 2:

1. “The literature review is limited, neglecting previous studies on Burnout in Portuguese Student Populations, including a recent (2021) study conducted with the same study population (medical students at the Faculdade de Medicina, University of Lisbon). The adaptation and renaming of the Burnout dimensions for the student population should be briefly explained. It is important to note that Student Burnout is not the same as general burnout in aid professions. The same applies to the selection of other variables. The rationale behind their inclusion (e.g., Social Support or Mental Health) and the evidence that supports their role as precursors of burnout should be provided. Additionally, it should be acknowledged that there are two medical schools in Lisbon.” 

Response: The study (2021) suggested by Reviewer 2 has been included in this new version. Despite the tight word limit, we attempted to clarify the rationale behind the introduction of the variables. As recommended, we have included the specific name of the faculty. The official english version of Faculdade de Medicina da Universidade de Lisboa is Lisbon School of Medicine, whilst Faculdade de Ciências Médicas da Universidade Nova de Lisboa goes by the designation NOVA Medical School. We have opted to change the name to the Portuguese name, making the distinction between the medical schools clearer.

5. “It is important to clarify the cutoff values for the Burnout scores that were considered indicative of Burnout. How were the Burnout scores calculated? Who validated the Portuguese version of the Maslach Burnout Inventory – Student Survey? What evidence of validity and reliability exists for the MBI-SS in the Portuguese population? The same questions apply to the other scales used. The manuscript lacks important information in this regard.”

Response: The new version includes the validation study of MBI-SS for the Portuguese population, as well as the study explaining how the cutoff values were defined. We have added the Portuguese validation studies of the instruments, along with the Cronbach’s alpha values for the dimensions of the instruments in our sample. The Portuguese version of the Maslach Burnout Inventory was shown to be a reliable valid instrument for the evaluation of the Burnout syndrome in Portuguese college students. The name of the Burnout dimensions for the student population were those of the aforementioned student survey. We also added information about the validation studies for the Portuguese population of the remaining scales and the Cronbach’s alpha values in our sample.

2. “The psychometric properties of the latent measures used (Burnout, Social Support, Mental Health) are not adequately described. Were the data collected in the present sample gathered using valid and reliable scales?” 

Response: The scales used are validated for the Portuguese population. The studies of their respective validations are indicated in the description of each instrument. Additionally, the Cronbach’s alpha values for each dimension of the scales used have been added, all of which show a good internal consistency.

3. “The major study variables are latent, so Structural Equation Modelling should be employed instead of classical repeated measures ANOVA and correlational analysis. It is also advisable for the authors to reconsider the use of the “A-NOVA” abbreviation, as it is not commonly used. With three time points, I would recommend using latent growth models.”

Response: We appreciate the identification of the abbreviation error for ANOVA, which has been corrected in this new version. While we acknowledge that Structural Equation Modelling (SEM) would be the ideal approach, it’s important to note that the paired sample size in our study (N = 108) falls below the generally recommended sample size criteria for SEM. In the literature, there is no consensus on the exact minimum sample size for SEM, but it is typically suggested to be around N=100-150 (Tinsley and Tinsley, 1987; Anderson and Gerbing, 1988; Ding, Velicer, and Harlow, 1995; Tabachnick and Fidell, 2001)Some researchers even advocate for larger sample sizes, such as N=200 (Hoogland and Boomsma, 1998; Boomsma and Hoogland, 2001; Kline, 2005).On the other hand, there is no minimum sample size for the use of the Generalised Linear Model (GLM). Due to this limitation in our sample size, we opted for the GLM as a statistical solution.

4. “The sample used in the study is not representative, as only 5% of the study population was selected, and the selection was not random.”

Response: – The percentage mentioned represents the proportion of students who participated in all three phases. This specific group allows us to analyse the evolution of burnout over time. We regret that it was not possible to pair all participants in the three phases. However, given the scarcity of prospective longitudinal studies in this research area, we consider it a strong point of the study and complementary to the more in-depth assessment in phase 3, where a higher percentage of burnout was observed. Additionally, for the three unpaired samples (P1 n = 443; P2 n =403; P3 n =332), with a 95% confidence level and a 5% margin of error, the recommended sample size is 327. This study was conducted among all medical students using an online Google Form that did not record email addresses. It was distributed to all medical students via institutional email. Any limitations related to the percentage of the paired sample are transparently disclosed in the limitations section for full transparency. 

6. “Provide standard errors (SE) for the data points in Figure 1.”

Response: As recommended, the errors have been incorporated into the Notes of Figure 1.